# Changes in Proximal Tubular Reabsorption Modulate Microvascular Regulation via the TGF System

**DOI:** 10.3390/ijms231911203

**Published:** 2022-09-23

**Authors:** Shayan Poursharif, Shereen Hamza, Branko Braam

**Affiliations:** 1Department of Medicine, Division of Nephrology and Immunology, University of Alberta, Edmonton, AB T6G 2G3, Canada; 2Department of Physiology, University of Alberta, Edmonton, AB T6G 2H7, Canada

**Keywords:** tubuloglomerular feedback, SGLT2 inhibitors, furosemide, diabetic nephropathy, acetazolamide, nephrovascular unit, renal microcirculation, synchronization, oxygenation–perfusion

## Abstract

This review paper considers the consequences of modulating tubular reabsorption proximal to the macula densa by sodium–glucose co-transporter 2 (SGLT2) inhibitors, acetazolamide, and furosemide in states of glomerular hyperfiltration. SGLT2 inhibitors improve renal function in early and advanced diabetic nephropathy by decreasing the glomerular filtration rate (GFR), presumably by activating the tubuloglomerular feedback (TGF) mechanism. Central in this paper is that the renoprotective effects of SGLT2 inhibitors in diabetic nephropathy can only be partially explained by TGF activation, and there are alternative explanations. The sustained activation of TGF leans on two prerequisites: no or only partial adaptation should occur in reabsorption proximal to macula densa, and no or only partial adaptation should occur in the TGF response. The main proximal tubular and loop of Henle sodium transporters are sodium–hydrogen exchanger 3 (NHE3), SGLT2, and the Na-K-2Cl co-transporter (NKCC2). SGLT2 inhibitors, acetazolamide, and furosemide are the most important compounds; inhibiting these transporters would decrease sodium reabsorption upstream of the macula densa and increase TGF activity. This could directly or indirectly affect TGF responsiveness, which could oppose sustained TGF activation. Only SGLT2 inhibitors can sustainably activate the TGF as there is only partial compensation in tubular reabsorption and TGF response. SGLT2 inhibitors have been shown to preserve GFR in both early and advanced diabetic nephropathy. Other than for early diabetic nephropathy, a solid physiological basis for these effects in advanced nephropathy is lacking. In addition, TGF has hardly been studied in humans, and therefore this role of TGF remains elusive. This review also considers alternative explanations for the renoprotective effects of SGLT2 inhibitors in diabetic patients such as the enhancement of microvascular network function. Furthermore, combination use of SGLT2 inhibitors and angiotensin-converting enzyme inhibitors (ACEi) or angiotensin receptor blockers (ARBs). in diabetes can decrease inflammatory pathways, improve renal oxygenation, and delay the progression of diabetic nephropathy.

## 1. Introduction

Recent studies strongly support that sodium–glucose co-transporter type 2 (SGLT2) inhibitors decrease cardiovascular risk [1,2]. In addition, these drugs preserve renal function, presumably by enhancing tubuloglomerular feedback (TGF) [3,4,5,6]. In this paper, we challenge this assumption. There is a widespread belief that SGLT2 inhibition causes a sustained increase in macula densa delivery leading to a sustained activation of TGF, which would then sustainably lower glomerular capillary pressure (PGC). However, there are two prerequisites for this proposition. First, adaptations in reabsorption upstream of the macula densa should not offset the original change in reabsorption. Second, no compensation should occur in the TGF mechanism that could offset a sustained TGF response. In this review, we extend our focus beyond SGLT2 inhibitors and consider several substances and conditions regarding these two prerequisites to help elucidate how these drugs may confer cardiovascular protection in situations where there is hyperfiltering states such as chronic kidney disease (CKD).

SGLT2 inhibitor, acetazolamide, and furosemide are compounds that can increase macula densa delivery and activate the TGF response, and can possibly decrease hyperfiltration. However, SGLT2 inhibitors seem to be the prototype of drugs to increase macula densa solute delivery and activate TGF response sustainably without changing TGF responsiveness. In contrast, furosemide and acetazolamide have characteristics that possibly fail to meet the prerequisites and prevent sustained increases in TGF activation. Besides the effect of single drugs, we will explore the effect of SGLT2 inhibitors in the real-life context of multiple medications for the treatment of diabetes mellitus and cardiovascular disease. An example would be the combined use of SGLT2 inhibitors and angiotensin-converting enzyme inhibitors (ACEi) or angiotensin receptor blockers (ARBs). These latter drugs would supposedly decrease TGF responsiveness, yet the large clinical trials showing the beneficial effects of SGLT2 inhibitors on renal function were performed in study populations where ACEi and ARB use was highly prevalent [7].

Another aspect of this is that while SGLT2 inhibitors were originally reported to specifically diminish the diabetic hyperfiltration of early diabetic nephropathy before anatomical injury to the glomerulus, they also exert preventative properties in more advanced diabetic nephropathy with segmental or global glomerulosclerosis. We will review the proximal tubular reabsorption and TGF dynamics in early versus more advanced diabetic nephropathy. Taken together, given what we know about SGLT2 inhibitors, TGF activation might be a component of the beneficial renal effects of SGLT2 inhibitors in diabetic nephropathy, but other mechanisms are likely and will be discussed.

## 2. Autoregulation

Kidneys stabilize RBF (renal blood flow), GFR, and PGC in parallel in response to fluctuations in blood pressure by adjustments in afferent arteriolar resistance by employing two mechanisms: the rapid myogenic response (MR) and the slower TGF. This autoregulation process stabilizes renal function and protects the glomeruli from injury [8]. 

MR is a universal response in all microvascular beds: a rise in wall tension, generally due to an increase in blood pressure, results in arteriolar vasoconstriction to stabilize tissue perfusion [9,10,11,12,13,14]. In the kidneys, the transmission of an increase in arterial pressure to the renal microvasculature leads to afferent arteriolar vasoconstriction, increased renal preglomerular resistance, and the stabilization of RBF, PGC, and GFR [8,14]. Calcium channel blockers can impair MR and decrease the gain of RBF autoregulation [11,15]. TGF is the second mechanism participating in RBF autoregulation [8,9,10,16]. Any residual increase in PGC and single nephron GFR (SNGFR) after compensation by MR will lead to an increase in the filtered load. Although a higher filtered load increases solute reabsorption in the proximal tubule and loop of Henle (glomerulotubular balance) [17], this still leads to an increase in solute delivery to the macula densa. This is sensed by the macula densa and results in the release of adenosine and adenosine triphosphate (ATP), which will lead to the constriction of the afferent arteriole of the same nephron. This, in turn, reduces single nephron GFR and RBF back to baseline, and eventually stabilizes the solute delivery to the macula densa [13,18,19,20]. Inhibiting TGF reduces RBF autoregulation efficacy, but does not abolish it [8,18]. TGF saturation is defined as the macula densa delivery, where any further increase would not result in any further activation of TGF and, consequently, a further decrease in SNGFR. Full TGF deactivation is defined as the macula densa delivery at which there is no influence of TGF on SNGFR. TGF responsiveness is defined as the maximum decrease in SNGFR elicited by the full activation of TGF. A sustained TGF response means that the depression of SNGFR by TGF activation upon a change in macula densa delivery remains the same over time. TGF blockade is the situation where a high concentration of furosemide is fully blocking how the macula densa can sense sodium, which is via the NKCC2 channel. 

MR and TGF interact synergistically [14,21]. It was demonstrated that with the influence of TGF, the afferent arteriole has a lower diameter and a further decrease in afferent arteriole diameter due to MR being markedly enhanced [14,17,21]. MR, in combination with TGF, is needed for an optimal RBF autoregulation and MR with its higher frequency acts as a dampening system, whereas TGF with its lower frequency modulates MR and represents a fine-tuning stabilization of renal function [22].

As mentioned in the introduction, compounds that decrease solute reabsorption upstream of the macula densa can sustainably increase macula densa delivery. However, partial compensation in reabsorption in segments proximal to the macula densa or resetting or saturation of the TGF system could offset this initial response. The overall thought is depicted in Figure 1.

## 3. Tubular Solute Reabsorption Proximal to the Macula Densa 

The reabsorption of sodium together with other solutes such as glucose, amino acids, chloride, and bicarbonate in the proximal tubule and loop of Henle ultimately determines the solute concentrations of fluid reaching the macula densa and the activation of TGF. In this section, we will further discuss the main sodium transporters in this context—NHE3, SGLT2, and the Na-K-2Cl co-transporter, NKCC2—and how they are regulated.

The three isoforms of the sodium–hydrogen exchanger (NHE1, NHE2, and NHE3) in the proximal tubule and descending and ascending loop of Henle [23,24] are mainly responsible for the reabsorption of sodium and bicarbonate. In a series of micropuncture studies in male Wistar rats, luminal amiloride, which inhibits NHE, significantly decreased NaCl and bicarbonate reabsorption in the proximal tubule [25,26]. Of the NHEs, the inhibition of NHE3 in the superficial nephrons of anesthetized rats reduced sodium and fluid reabsorption markedly [27]. Consequently, it is suggested that NHE3 is responsible for the reabsorption of a substantial proportion of sodium and bicarbonate in S1 and S2 segments of the proximal tubule, with concomitant paracellular water and chloride reabsorption [5].

NHE3 is regulated by several stimulatory and inhibitory hormones at transcriptional and post-translational levels, including changes in protein phosphorylation and trafficking [28]. Insulin and glucocorticoids increased NHE3 activity by increasing NHE3 transcription in opossum kidney cells [29,30]. Parathyroid hormone (PTH) decreased NHE3 activity in opossum kidney cells by phosphorylating NHE3 at various serine sites and decreasing surface NHE3 [31]. Angiotensin II (ANG II) increased NHE3 function by changing NHE3 phosphorylation [32,33,34]. In addition, it has been demonstrated that NHE3 and SGLT2 interact functionally in the proximal tubule as the inhibition of SGLT activity in rats by both empagliflozin and phlorizin significantly decreased NHE3 activity [35,36]. As such, compounds and hormones that directly or indirectly affect NHE3 regulation and activity can markedly affect the solute concentration of fluid reaching the macula densa and might affect other transporters.

Under physiological conditions, the sodium–glucose co-transporters (SGLTs) SGLT1 and SGLT2 in the proximal tubule reabsorb almost all of the filtered glucose [5]. This is achieved by the low-affinity/high-capacity SGLT2 in the early proximal tubule and the high-affinity/low-capacity SGLT1 in the late proximal tubule [6,37]. SGLT2 reabsorbs sodium and glucose on a 1:1 ratio, while SGLT1 reabsorbs two sodium ions along with one glucose molecule [38]. Moreover, like NHE3, SGLT2 participates in paracellular Cl and water reabsorption by creating an osmotic gradient [5].

SGLT2 is upregulated at the transcriptional and post-translational level by ANG II, insulin, and hepatocyte nuclear factor (HNF-1α) [39,40]. The increased transcription of the transcription factor HNF-1α in diabetic rats was associated with increased binding of HNF-1α into the SGLT2 promoter region, which then likely caused increased SGLT2 expression [41]. Additionally, the phosphorylation of SGLT2 by activating protein kinase A and C in human embryonic kidney 293T cells increased the reabsorption of sodium and glucose markedly [42]. 

In the loop of Henle, 25–30% of filtered Na-Cl is reabsorbed via NKCC2 across the apical membrane of the water impermeable thick ascending limb (TAL) [43,44]. Ninety percent of reabsorbed potassium via NKCC2 is secreted back into the lumen through inward-rectifying K channels in the apical membrane. The secretion of potassium to lumen and the absorption of chloride from the basolateral membrane to the circulation via chloride channels generates a lumen-positive transepithelial potential difference. This mediates the paracellular reabsorption of sodium, magnesium, and calcium [44,45,46]. Unlike TAL, the thin limb of Henle generally has low permeability to sodium and does not contribute significantly to sodium reabsorption [47,48,49]. 

Several hormones modulate the NaCl reabsorption in TAL by regulating NKCC2 activity. NKCC2 is stimulated at the post-translational level mainly via increasing intracellular cyclic adenosine monophosphate (cAMP) levels by stimulatory hormones such as vasopressin, PTH, and calcitonin. cAMP enhances NKCC2 activity by stimulation insertion into the cell membrane and presumably also by phosphorylation [43]. The stimulation of NKCC2 by phosphorylation was shown in cultured TAL cells from rabbits and in rats with diabetes insipidus after administrating vasopressin agonists [50]. Conversely, nitric oxide (NO), atrial natriuretic peptides (ANP), endothelin, and prostaglandin E2 (PGE2) inhibit NKCC2 mainly via cyclic guanosine monophosphate (cGMP) by decreasing its trafficking [43,44]. It was also shown that Ang II increased NaCl transport in rats TAL by blunting the NO-induced inhibition of NKCC2, which increased NKCC2 activity [51,52]. Furthermore, the administration of indomethacin and diclofenac, prostaglandin inhibitors, in rats decreased solute reabsorption in the loop of Henle by upregulating NKCC2, while misoprostol, a PGE2 analog, reset this effect [53]. From the above, it is clear that NKCC2 activity is modulated by many factors.

## 4. Effects of Early and Late Diabetic Nephropathy on Macula Densa Solute Delivery and the TGF Response

In the early phase of diabetic nephropathy, proximal tubular reabsorption is increased, resulting in decreased solute delivery to the macula densa and deactivation of the TGF response [54], hyperfiltration, and the progression of glomerular damage. This, in turn, could change solute reabsorption upstream of the macula densa and alter the TGF response in a variety of ways. We will discuss how macula densa solute delivery and the TGF system are affected in early and advanced diabetic nephropathy (Figure 2).

In diabetes mellitus, the glomerular-filtered load of glucose is increased. Nephrons become hypertrophied and SGLT2 expression increases in the proximal tubule [6]. Increased reabsorption by SGLT2s would result in the decreased solute concentration of fluid reaching the macula densa, deactivation of TGF, and hyperfiltration. That hyperfiltration was a response to increased transport by SGLT2 was shown in streptozotocin-induced type 2 diabetic mice and Akita mouse models of type 1 diabetes: hyperfiltration did not occur if diabetes was induced in SGLT2 knockout mice [56]. Additionally, in early diabetes, the turning point of the TGF curve can be shifted upward (TGF is reset upward), which can exacerbate the hyperfiltration [57]. Furthermore, plasma and kidney ANG II levels are increased in diabetes mellitus [58]. This was also shown in several type 1 and type 2 animal models as derived from increased angiotensinogen levels [59,60,61]. Increased ANG II levels in diabetes can modulate the TGF system in several ways. As mentioned, ANG II enhances solute reabsorption in the proximal tubule and loop of Henle. Therefore, macula densa solute delivery decreases and TGF is deactivated, which could exacerbate hyperfiltration [62,63]. However, ANG II resets the turning point of the TGF curve to a lower level so that, potentially, SNGFR does not change in response to increased ANG II [64]. Taken together, in early diabetic nephropathy, increased solute reabsorption by SGLT2s would result in hyperfiltration, while increased ANG II might exacerbate hyperfiltration slightly.

In advanced diabetic nephropathy, nephrons can display at least three different phenotypes: intact nephrons, segmentally sclerosed nephrons, and globally sclerosed nephrons. The intact, often hypertrophied nephrons might display enhanced reabsorption due to the same mechanisms as in early diabetes. Increased ANG II levels and the increased activity of SGLT2 transporters will contribute to deactivation of the TGF and cause hyperfiltration. 

Additionally, some nephrons can become segmentally sclerosed, which can result in decreased filtration, decreased macula densa delivery, and the deactivation of TGF. In these partially sclerosed nephrons, GFR might or might not be increased based on the extent of glomerular sclerosis. If there is an adequate glomerular surface, GFR would increase. Otherwise, GFR would remain low.

Finally, some nephrons are globally sclerosed and there is no filtration and TGF activity. This would contribute to an overall decrease in GFR. These globally sclerosed nephrons no longer consume oxygen. As further explained below, such nephrons can also disturb microvascular network dynamics in two ways. First, they can cause anatomical disruption. Second, they disrupt the oxygenation of nearby intact nephrons, which can result in decreased solute reabsorption, increased macula densa solute delivery, the activation of TGF, and a decrease in SNGFR in nearby intact nephrons.

Additionally, volume expansion in advanced diabetic nephropathy might affect macula densa delivery and the TGF system. Volume expansion decreases ANGII levels [65]. This would lead to decreased reabsorption and TGF responsiveness [66] in the intact and partially sclerosed nephrons, which would result in decreased GFR.

In summary, in early diabetic nephropathy, the solute concentration of fluid reaching the macula densa decreases and TGF is deactivated, which would result in hyperfiltration. In advanced diabetic nephropathy, the changes in reabsorption and the TGF response are difficult to predict due to segmental or global glomerulosclerosis and potential volume expansion (Figure 2). 

## 5. Effects of SGLT2 Inhibitors on Macula Densa Solute Delivery and TGF Response

Clinically, SGLT2 inhibitors have been shown to be effective in reducing cardiovascular events and in preventing CKD progression in mild, but also more advanced, diabetic as well as non-diabetic CKD [1,2,67]. SGLT2 inhibitors directly and indirectly (via NHE3) inhibit the reabsorption of sodium in the proximal tubule, resulting in an immediate increase in solute delivery to the macula densa [36,47]. Chronically, this is partially offset by an increase in the function of SGLT1 in the proximal tubule as well as an increase in loop of Henle solute reabsorption [4]. In a series of micropuncture studies assessing acute and chronic effects of the SGLT2 inhibitor, dapagliflozin, on proximal tubular reabsorption in streptozotocin-induced diabetic rats, it was concluded that the inhibitory effect of dapagliflozin on proximal reabsorption is partially compensated by an increase in the loop of Henle solute reabsorption [4]. Additionally, the inhibition of SGLT2 in mice showed an increase in glucose reabsorption by SGLT1 from 3% to 50–60% of total glucose reabsorption [68]. Besides this, the turning point of the TGF curve can be shifted during chronic SGLT2 inhibition. An increase in the solute concentration of fluid reaching the macula densa in the mouse renal cortex can lead to the generation of nitric oxide [69]. This has been shown to offset the TGF response in mice and rats [70,71,72]. 

Nevertheless, SGLT2 inhibitors can sustainably increase the solute concentration of fluid reaching the macula densa, and lead to a sustained TGF activation, afferent arteriolar vasoconstriction, and a decline in SNGFR and whole-kidney GFR. However, in the only available study on this subject, the sustained response was 50% of the acute response [4,5,6]. 

Besides the different phases of diabetic and non-diabetic CKD, an area of great uncertainty is whether other concomitant treatments could offset the beneficial effects of SGLT2 inhibitors. An example is ACE inhibitor therapy. It has been documented by us [73,74] and others [75,76,77] that the acute administration of ACE inhibitors decreases the TGF maximum response and increases the tubular fluid flow, exerting a half-maximum TGF response in experimental animals. Nevertheless, we are only aware of one study from our own group demonstrating that after prolonged ACE inhibition, the TGF response returns [78]. Whether a TGF response is present and of normal magnitude in humans during prolonged ACE inhibition is entirely unknown. Similar considerations would apply to intensive treatment with calcium channel blockers and loop diuretics; these medications could also modify TGF responsiveness and modulate the response to SGLT2 inhibition, but information in both experimental settings and in humans is lacking. 

The available literature about the mechanistic aspects of the positive effects of SGLT2 inhibitors in early diabetic nephropathy is based on a relatively small number of studies in rats and mice supporting the idea that these compounds increase the solute concentration of fluid reaching the macula densa sustainably and activate TGF sustainably, reducing hyperfiltration and thereby preserving renal function. The literature is lacking a solid basis for how SGLT2 inhibitors exert their actions in more advanced diabetic and non-diabetic CKD. In addition, TGF is difficult to assess and has hardly been studied in humans, and whether TGF is the central mediator of positive effects of SGLT2 inhibitors in human diabetic and non-diabetic CKD patients remains elusive. 

## 6. Effects of ACTZ on Macula Densa Solute Delivery and TGF Response

Acetazolamide reduces proximal tubular reabsorption by inhibiting bicarbonate reabsorption via decreasing carbonic anhydrase activity and NHE3 [79]. This initially increases macula densa solute delivery. Despite a compensatory increase in proximal tubular and loop of Henle solute reabsorption due to an increase in ANG II levels [80], acetazolamide can increase the solute concentration of fluid reaching the macula densa sustainably, and initially reduces GFR by activating TGF and potentially increasing the proximal tubular pressure [81,82]. However, TGF does not remain activated due to the upward and rightward resetting of the turning point of the TGF curve, possibly due to increased NO levels [83,84,85]. In a study assessing autoregulation, increased renal plasma pressure resulted in an immediate afferent arteriolar vasoconstriction followed by a moderate decline of afferent arteriolar vasoconstriction (sustained response). Acetazolamide administration enhanced the initial vasoconstriction, but it did not alter the sustained response. The inhibition of NO enhanced both the initial and sustained constrictor response to acetazolamide [86]. Consequently, acetazolamide did not sustainably change single nephron and whole-kidney hemodynamics due to the compensation occurring in the TGF response (TGF resetting). Of note, acetazolamide would have been an interesting control in studies about SGLT2 inhibitors since it sustainably decreases proximal tubular reabsorption, but does not sustainably activate TGF, and both compounds are diuretics. 

## 7. Effects of Furosemide on Macula Densa Solute Delivery and TGF Response

Furosemide reduces sodium reabsorption in the thick ascending limb of the loop of Henle by inhibiting NKCC2, which would initially increase the solute concentration of fluid reaching the macula densa. While this can be compensated by increasing the solute reabsorption in the proximal tubule due to the activation of renin-angiotensin system (RAS) and the sympathetic nervous system, furosemide can increase the solute concentration of fluid reaching the macula densa sustainably [87]. That said, the effect of furosemide on the TGF system is less clear. One scenario is that furosemide would activate and then saturate the TGF response, effectively eliminating TGF dynamics on renal autoregulation. Another option is that furosemide can (partially) inhibit the sensing step of the TGF system since this is formed by NKCC2 in the macula densa. This would also prevent a sustained activation of the TGF response. It is suggested that the tubular concentration of furosemide is the determinant of the balance between these two opposite responses [87]. Consequently, while furosemide can sustainably increase the solute concentration of fluid reaching the macula densa, it is not clear whether it can sustainably alter TGF activation.

## 8. Alternative Explanations for the Beneficial Effects of SGLT-2 Inhibitors in Diabetic Nephropathy

Although the data in the literature are sparse, there are other explanations for the positive effects of SGLT2 inhibitors in diabetic nephropathy. SGLT2 inhibitors could enhance the network function among nephrons by activating the TGF system. In addition, the combination use of SGLT2 inhibitors and ARBs/ACEi can reduce fibrosis, inflammatory pathways, and glomerular injury in diabetes and thereby preserve renal function. Furthermore, SGLT2 inhibitors and ARBs/ACEi decrease tubular reabsorption. This reduces oxygen demand and renal hypoxia, which are known cause of fibrosis and microvascular injury [88].

Recognition of the complexity of the kidney’s microvascular physiology and anatomy has altered the idea that each nephron is responsible for its own autoregulation. We and others have suggested that nephrovascular units (NVUs), which consist of a nephron and its afferent and efferent arteriole, communicate with each other in a network, and explain renal autoregulation and perfusion in the kidneys better [89]. TGF generates oscillations within each NVU, and oscillatory systems with similar frequencies can entrain and become synchronized [90,91,92]. This synchronization among NVUs, which increases RBF autoregulation efficacy, could prevent a discrepancy between metabolic demand and oxygen delivery as each NVU’s tubule is perfused by four or five nearby NVUs [89]. Consequently, one theory explaining the renoprotective effects of SGLT2 inhibitors is that they can enhance network dynamics among NVUs to activate the TGF response sustainably.

Moreover, TGF-generated oscillations can be transmitted to upstream vascular branch points through endothelial gap junctions formed by connexin proteins (mainly connexin 40) [89]. This upstream “electrical cable” enables communication between NVUs at vascular branch points and optimizes network among them. It was shown that treating mice with the metabolic syndrome with SGLT2 inhibitors prevented the decline in connexin 40 in these animals [93]. Consequently, SGLT2 inhibitors can also directly affect the transmission of TGF oscillations to upstream microvasculature, and thereby enhance network function amongst them.

Another explanation for the beneficial effects of SGLT2 inhibitors could be the combined use of SGLT2 inhibitors and ACEi or ARBs in diabetic patients. Increased luminal glucose by SGLT2 inhibitors inhibits urate transporter 1, which would increase uric acid excretion substantially. This in turn would reduce reactive oxygen species (ROS), inflammation, and renal damage induced by uric acid [70]. SGLT2 inhibitors can also reduce the mRNA expression of proinflammatory mediators such as nuclear factor-κB and interleukin 6 (IL-6) levels in the kidney [94,95]. Moreover, it was shown that ACEi can prevent tubulointerstitial fibrosis, tubular apoptosis, and renal oxidative stress in animal studies [96,97]. Furthermore, the combined use of SGLT2 inhibitors and ACE inhibitors reduces tubular reabsorption additively, which can further decrease hyperfiltration. Moreover, this reduces oxygen demand and would improve renal oxygen levels [88]. Consequently, a combination of SGLT2 inhibitors and ARBs/ACEi could have synergistic effects to decrease inflammatory pathways, improve renal oxygenation, and delay the progression of diabetic nephropathy. Nevertheless, ambiguity remains about the combination of SGLT2 and ARBs/ACEi, since acute administration of ARBs/ACEi decreases TGF responsiveness in rats and thereby could attenuate the TGF activation by SGLT2 inhibitors [78]. There are no data available about TGF responses in humans, and no accurate method is currently available for assessment. Further studies to elucidate the mechanism on the effects of SGLT2 inhibitors on diabetic, but also non-diabetic, nephropathy are clearly needed.

## 9. Conclusions

In summary, two prerequisites are necessary to make a compound effective to sustainably activate the TGF system and sustainably alter renal hemodynamics. First, macula densa delivery should increase sustainably without any significant compensation in reabsorption in the segments upstream of the macula densa. Second, no compensation should occur in the TGF response. SGLT2 inhibitors seem to be an example of compounds that can activate the TGF system sustainably, which is one of the explanations for the positive effects of SGLT2 inhibitors in advanced diabetic nephropathy, as they decrease glomerular hyperfiltration by sustainably activating the TGF system. Moreover, SGLT2 inhibitors can enhance network dynamics among NVUs and decrease ROS, which can delay the progression of diabetic nephropathy. Finally, in diabetic patients, the combination use of SGLT2 inhibitors and ARBs/ACEi could contribute to the enhancement of renal hemodynamics. However, many questions remain regarding the exact mechanism of action of SGLT2 inhibitors.

## Figures and Tables

**Figure 1 ijms-23-11203-f001:**
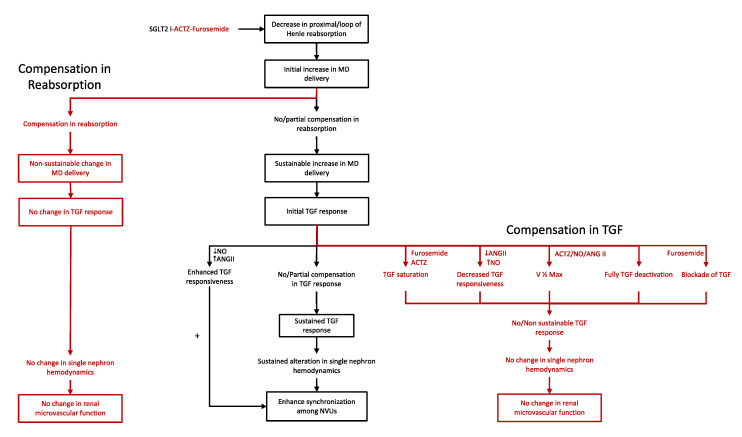
Compounds and conditions that sustainably activate the tubuloglomerular feedback (TGF) mechanism. Compounds that decrease solute reabsorption in the proximal tubule/loop of Henle can sustainably increase macula densa (MD) delivery unless there is a compensation in the reabsorption in segments proximal to the macula densa (**left**). Sustained increase in macula densa delivery would lead to sustained activation of the TGF system unless there is a compensation in the TGF response (**right**). Sustained activation of the TGF mechanism would result in sustained alteration of single nephron hemodynamics and presumable enhancement of synchronization among nephrovascular units (NVUs) (**center**).

**Figure 2 ijms-23-11203-f002:**
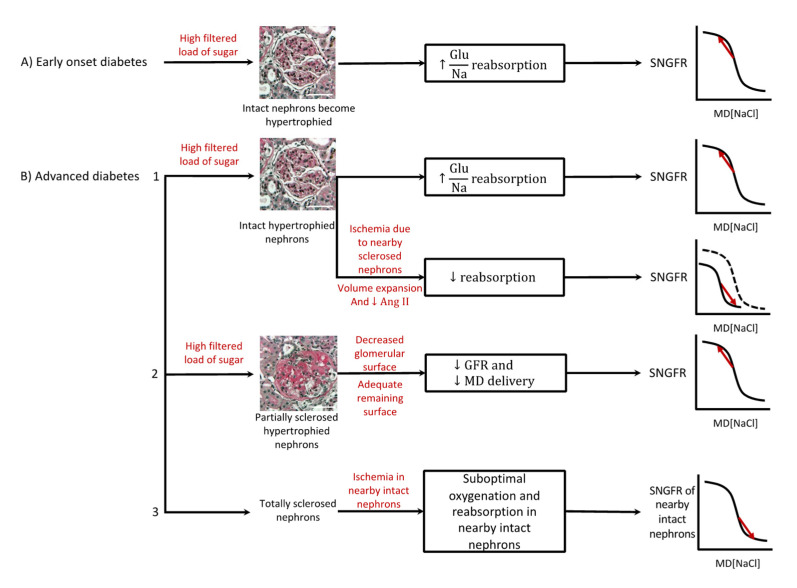
Macula densa (MD) solute delivery and the tubuloglomerular feedback (TGF) response in early and advanced diabetic nephropathy. (**A**) Shows intact, hypertrophied nephrons in early diabetic nephropathy, which would result in increased solute reabsorption proximal to the macula densa and deactivation of the TGF system and hyperfiltration. (**B**_1_) Shows intact hypertrophied nephrons in advanced diabetic nephropathy and how the macula densa salute delivery and the TGF response might be changed based the extent of sclerosis in nearby nephrons (dashed curve shows normal TGF curve). (**B**_2_) Shows focally sclerosed, hypertrophied nephrons in advanced diabetic nephropathy, which would result in decreased GFR and macula densa salute delivery and deactivation of the TGF system. (**B**_3_) Shows globally sclerosed nephrons and how they might affect nearby intact nephrons [55]. Scale bars indicate 50 μm.

## Data Availability

Not applicable.

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
