# Peer review of "Changes in Proximal Tubular Reabsorption Modulate Microvascular Regulation via the TGF System"

_ijms, 2022, doi:10.3390/ijms231911203_

Round 1

Reviewer 1 Report

The authors offer an up-to-date review on how sodium-glucose cotransporter 2 inhibitors preserve renal function by decreasing proximal tubular reabsorption and activating the tubuloglomerular feedback. The authors discuss compounds and conditions that activate the TGF mechanism under the influence of different inhibitors with different mechanisms of action: SGLT2 inhibitors, acetazolamide, and furosemide that target sodium-dependent glucose cotransporters, sodium-hydrogen exchanger 3,  and the Na-K-2Cl co-transporter, respectively. The authors explore possible changes in TGF efficiency in early and advanced diabetic nephropathy.

Comments and suggestions:

1) The authors should explain the terminology they use: "TGF responsiveness" and "Sustained TGF activation" . These terms are not widely used and very confusing. For example, does TGF responsiveness depend on both the slope and min-max range of the S-curve?

2) Fig. 1 contains terms "TGF saturation", "Decreased TGF responsiveness", "Fully TGF  deactivation", "Blockage of TGF" . The meaning of these terms is not obvious. I suggest being pedagogical and explaining them. I found "TGF saturation" being irrelevant for furosemide (see 7) below).

3) Fig. 2 does not contain A, B1, B2, B3 but the figure caption discusses them.4) What is the dashed curve in Fig. 2? It is not described in the figure caption.

5) The authors do not discuss much  TGF resetting which is an important issue addressed e.g Aihua Deng, John S Hammes, Scott C Thomson,  Kidney international 62(6), pp. 2136–2143 (2002) among others.

6)  The authors suggest alternative explanations for the beneficial effects of SGLT2 inhibitors in CKD focusing on the entire renal microvasculature. I found their statements being speculative and not supported by data.

7) Section 7. "Effects of furosemide on macula densa solute delivery and TGF response". It is widely accepted that furosemide blocks NKCC2 in the macula densa such that it deactivates TGF. The author's discussion about saturation of the TGF response seems to be irrelevant. 

Author Response

Thank you for your comments. For the sake of easiness, we included your comments and suggestions as well:

The authors offer an up-to-date review on how sodium-glucose cotransporter 2 inhibitors preserve renal function by decreasing proximal tubular reabsorption and activating the tubuloglomerular feedback. The authors discuss compounds and conditions that activate the TGF mechanism under the influence of different inhibitors with different mechanisms of action: SGLT2 inhibitors, acetazolamide, and furosemide that target sodium-dependent glucose cotransporters, sodium-hydrogen exchanger 3, and the Na-K-2Cl co-transporter, respectively. The authors explore possible changes in TGF efficiency in early and advanced diabetic nephropathy.

Comments and suggestions:

1) The authors should explain the terminology they use: "TGF responsiveness" and "Sustained TGF activation”. These terms are not widely used and very confusing. For example, does TGF responsiveness depend on both the slope and min-max range of the S-curve?

Thank you for your comments. We have added a few lines with definitions regarding the TGF system (line 99 and onwards). This partly addresses your second comment also.

2) Fig. 1 contains terms "TGF saturation", "Decreased TGF responsiveness", "Fully TGF deactivation", "Blockage of TGF”. The meaning of these terms is not obvious. I suggest being pedagogical and explaining them. I found "TGF saturation" being irrelevant for furosemide (see 7) below).

Thanks for your comment. As mentioned, we have now included some sentences with definitions.

Although furosemide can inhibit TGF by inhibiting NKCC2 transporters at macula densa, it was suggested that high tubular concentration of furosemide is needed to fully block the NKCC2 transporters at macula densa. The amount of evidence is limited, yet furosemide in low concentrations is likely to increase macula densa delivery and not block TGF sensing, thus resulting in TGF activation and a decrease in GFR. In higher concentration, it might result in saturation of the TGF response, that is increasing macula densa delivery so, that it leads to maximum activation of TGF, with a more pronounced decrease in GFR. At very high concentrations, it could block the sodium sensing, and lead to an increase in GFR. In our view saturation is relevant to mention, since it prevents that the TGF system can dynamically participate in autoregulation: the response is ‘clamped’.

3) Fig. 2 does not contain A, B1, B2, B3 but the figure caption discusses them.4) What is the dashed curve in Fig. 2? It is not described in the figure caption.

Thanks for your comment. The figure and its caption were changed slightly.

5) The authors do not discuss much TGF resetting which is an important issue addressed e.g Aihua Deng, John S Hammes, Scott C Thomson, Kidney international 62(6), pp. 2136–2143 (2002) among others.

Thanks for your comment. Resetting of TGF turning point after acetazolamide was discussed briefly in line 294-295. Also, TGF resetting in diabetes was discussed in section 4. We made small changes to further highlight the important role of TGF resetting.

6)  The authors suggest alternative explanations for the beneficial effects of SGLT2 inhibitors in CKD focusing on the entire renal microvasculature. I found their statements being speculative and not supported by data.

The section’s idea really was to point at this gap in the literature. We have now explicitly mentioned this.

7) Section 7. "Effects of furosemide on macula densa solute delivery and TGF response". It is widely accepted that furosemide blocks NKCC2 in the macula densa such that it deactivates TGF. The author's discussion about saturation of the TGF response seems to be irrelevant. 

This was explained in response to your second comment.

Reviewer 2 Report

This review manuscript summarized the role of TGF-related mechanism that may be involved in renal protection of SGLT2 inhibitor in an innovative way compared with other review papers.  It is well written and may help clinics understand the effect of SGLT2 inhibitor for DKD or CKD, especially for a long-term use. 

Author Response

Thank you very much for reviewing our paper. We really appreciate your comments.

Reviewer 3 Report

Shayan Poursharif and colleagues submitted a manuscript entitled “Changes in proximal tubular reabsorption modulate microvascular regulation via the TGF system” for publication as a review in IJMS. The authors focused on role of SGLT2 inhibitors in the regulation of tubuloglomerular feedback (TGF) mechanism in addition to the beneficial effects under the condition of diabetic nephropathy. Individuals with DN, develop the hypertension and combination of SGLT2 inhibitors and ARB can have better outcomes in decreasing the progression of advanced DN.  The current review manuscript is well written, but several issues raise concerns.

1.     As review major focused on the SGLT2 inhibition regulating the TGF system in DKD, i would suggest to modify the title by stating the role of SGLT2 inhibition or blockage of angiotensin ii receptor modulate microvascular environment by regulating tubuloglomerular feedback mechanism.  

2.     Although authors mentioned about the effects of other diuretics (Acetazolamide and Furosemide) regulating the Na reabsorption, which may be out of focus or little over to discuss in this current review. Also, as these diuretics may partially control the TGF mechanism or not much evident.

3.     Delete the complete section 8 (Line 308) as authors focused on the SGLT2 inhibition in CKD which is much less evident or minimal studies available.

4.     Would be an addition if authors could discuss about the effect of RAAS blockage in regulating the TGF mechanism.  

5.     It would much easier if authors could add a Representing figure suggesting the TGF mechanism in different disease models of diabetes and hypertension and ischemic condition etc.

Author Response

Thank you for your comments. For the sake of easiness, we included your comments and suggestions as well:

Shayan Poursharif and colleagues submitted a manuscript entitled “Changes in proximal tubular reabsorption modulate microvascular regulation via the TGF system” for publication as a review in IJMS. The authors focused on role of SGLT2 inhibitors in the regulation of tubuloglomerular feedback (TGF) mechanism in addition to the beneficial effects under the condition of diabetic nephropathy. Individuals with DN, develop the hypertension and combination of SGLT2 inhibitors and ARB can have better outcomes in decreasing the progression of advanced DN.  The current review manuscript is well written, but several issues raise concerns.

  1. As review major focused on the SGLT2 inhibition regulating the TGF system in DKD, i would suggest to modify the title by stating the role of SGLT2 inhibition or blockage of angiotensin ii receptor modulate microvascular environment by regulating tubuloglomerular feedback mechanism.  

Thank you for your comments. It is unfortunate we did not make it clear enough that the paper is considering not only SGLT2 inhibitors, but other diuretics that could have similar effects on TGF (by increasing macula densa solute delivery) and possibly have some long-term renoprotective effects in hyper filtering states such as DKD. To avoid this confusion, we have added an opening sentence in the abstract to indicate this more clearly.

Moreover, our reason for reviewing the effects of ACEi/ARBs was to better explain the renoprotective effects of SGLT2 inhibitors in real life context where many patients take ACEi/ARBs, which are known for their renoprotective effects.

  1. Although authors mentioned about the effects of other diuretics (Acetazolamide and Furosemide) regulating the Na reabsorption, which may be out of focus or little over to discuss in this current review. Also, as these diuretics may partially control the TGF mechanism or not much evident. 

Sorry for the confusion. One of the main messages of this review paper is on two prerequisites a compound should have to possibly enhance renal hemodynamics in hyper filtering situations such as diabetic nephropathy. First, it should sustainably increase macula densa delivery and second it should activate the TGF response sustainably. This was the main reason that we reviewed the effects of SGLT2 inhibitors, acetazolamide and furosemide that can decrease proximal tubular reabsorption and can possibly activate the TGF and decrease hyperfiltration. We argued that only SGLT2 inhibitors can sustainably enhance macula densa solute delivery and activate the TGF. However, acetazolamide and furosemide cannot sustainably activate TGF due to compensation in reabsorption and TGF response. These diuretics have not been thoroughly studied and the long-term effects of these diuretics on diabetic nephropathy remains elusive. We made small changes in the introduction to make the reasons for reviewing acetazolamide and furosemide clearer.

  1. Delete the complete section 8 (Line 308) as authors focused on the SGLT2 inhibition in CKD which is much less evident or minimal studies available. 

It is unfortunate that we could not make it clear enough that we argued throughout the paper that positive effects of SGLT2 inhibitors in early and diabetic nephropathy can be only partially explained by activation of TGF. So, we reviewed other explanations for positive effects of SGLT2 inhibitors, which are explained in this section.  We made small changes in the title and first line of this section (line 321-324) to make the alternative explanations for preservative effects of SGLT2 inhibitors in diabetic nephropathy clearer.

  1. Would be an addition if authors could discuss about the effect of RAAS blockage in regulating the TGF mechanism.  

We clearly have not been comprehensive in explaining the effects of ANG II and RAAS blockade. Assessing the effects of ACEi/ARBs on TGF is very complex as they affect both reabsorption and TGF responsiveness. We have discussed the effects of ACEi/ARBs on reabsorption and TGF responsiveness in line 357-358 and 363-364 respectively. We also reviewed the possible effects of ANG II on reabsorption, TGF activity and responsiveness in section 4. We made changes in line 358-359 to emphasize more on the role of ACEi on tubular reabsorption and TGF response.

  1. It would much easier if authors could add a Representing figure suggesting the TGF mechanism in different disease models of diabetes and hypertension and ischemic condition etc. 

Sorry we have not been too explicit about this. Therefore, we would like to make this clearer that the paper predominantly focuses on situations where there is hyperfiltration state such as CKD, where the remaining intact nephrons hyper perform by hyperfiltration. This would eventually lead to glomerular injury. Beside the diabetic nephropathy that we discussed (figure 2), discussing hypertensive kidney injury, where there is microvascular damage, or ischemic nephropathy would be different situations. We have changed the introduction slightly to make sure readers understand that we mainly focused on hyperfiltration situations. We made small changes in the abstract (Line 12) and introduction (Line 48) to address your kind comment. Moreover, in figure 2, we tried to represent changes in MD solute delivery and TGF response in early and advanced diabetic nephropathy in different nephron’s states.

Round 2

Reviewer 3 Report

Thank you. All the comments and suggestions were addressed by authors. Revised version of current manuscript can be accepted for publication in IJMS.